# Investigating Different Local Polyurethane Coatings Degradation Effects and Corrosion Behaivors by *Talaromyces funiculosus* via Wire Beam Electrodes

**DOI:** 10.3390/ma16041402

**Published:** 2023-02-07

**Authors:** Xiangping Hao, Kexin Yang, Yiding Yuan, Dawei Zhang, Lin Lu

**Affiliations:** 1National Materials Corrosion and Protection Data Center, Institute for Advanced Materials and Technology, University of Science and Technology Beijing, Beijing 100083, China; 2BRI Southeast Asia Network for Corrosion and Protection (MOE), Shunde Innovation School, University of Science and Technology Beijing, Foshan 528399, China; 3Beijing Advanced Innovationation Center for Materials Genome Engineering, University of Science and Technology Beijing, Beijing 100083, China

**Keywords:** *T. funiculosus*, wire beam electrode, polyurethane coating, tropical rainforest environment, biodegradation

## Abstract

The degradation effect of mold on the coating in a hot and humid environment is one of the important factors that cause layer failure. Combined with the wire beam electrode (WBE) and the traditional surface analysis technique, the local biodegradation of the coatings and the corrosion behaviors of metal substrates can be characterized accurately by a WBE. Herein, a WBE was used to study the degradation impact of *Talaromyces funiculosus* (*T. funiculosus*) isolated from a tropical rainforest environment on the corrosion of polyurethane (PU) coating. After immersion for 14 days, the local current density distribution of the WBE surface can reach ~10^−3^ A/cm^2^ in the fungal liquid mediums but maintains ~10^−7^ A/cm^2^ in sterile liquid mediums. The |Z|_0.01Hz_ value of the high current densities area (#85 electrode) was 1.06 × 10^9^ Ω cm^2^ in a fungal liquid medium after 14 days of immersion. After being attacked by *T. funiculosus*, the degradation of the PU was more severe, and there were wrinkles, cracks, blisters, and even micro-holes distributed randomly on the surface of electrodes. This resulted from the self-corrosion caused by the *T. funiculosus* degradation of the coating; the corrosion caused by the electric coupling effect of the coating was introduced. Energy dispersive spectroscopy (EDS) and Raman spectra results showed that the corrosion products were flakey and globular, which consisted of γ-FeOOH, γ-Fe_2_O_3_, and α-FeOOH.

## 1. Introduction

Polyurethane (PU) coatings, as one of the most common organic coatings, are widely applied in the automobile industry, engineering construction, aerospace field, light textile industry, etc. because of their pliability, durability, versatility, and abrasive resistance [1,2,3,4]. Thanks to their barrier effect against corrosive media and their inhibition effect on corrosion reactions, PU coating is also an ideal candidate utilized in mitigating the corrosion of metals [5,6]. However, the factors of light, oxygen, microorganisms, temperature, and moisture can damage the cross-link structures of PU and accelerate the formation of blistering and chalking [7,8]. Among them, the first three factors dominantly contribute to the degradation of PU in natural environments [9,10]. When the PU coating surface is exposed to light, part of the energy from it is absorbed by the polymer backbone and side chains and triggers the rupture of chemical structures, which is the main cause of photoaging [11]. During this process, the C-O bond on the backbone structure could break and generate amino radicals and CO_2_ [11]. Moreover, the presence of oxygen can accelerate the photoaging process of the coatings and change this process into a photo-oxidation degradation process [12].

Compared with the chemical degradation process caused by light and oxygen, the degradation process caused by microorganisms, especially fungi, is a synergistic process of chemical degradation and biodegradation. Because multiple degradation pathways exist simultaneously, the degradation process of PU by microorganisms is complicated [13,14,15,16]. Khan et al. summarized the procedure of the biodegradation process of PU coatings caused by *Aspergillus tubingensis* [17]. First, the spores of fungi adhered to the PU coatings randomly, followed by growing into mycelium on the coating surface. During this process, PU, as a nutrition source, can supply energy for the metabolism of fungi. Metabolites such as esterase can hydrolyze the carbonyl groups with time, completing the biodegradation process through the metabolite action and the mechanical destruction of mycelium. 

The coating failure caused by fungi usually occurs locally because the colonization of fungus on coating surfaces is usually random. Hence, compared with the conventual electrochemical method, wire beam electrode (WBE) technology is a preferred way to estimate the localized failure of coatings and evaluate the local corrosion of metals [18]. The WBE is combined with many wires to simulate a whole large area of metal, and each metal wire can be separated from the others. Thereinto, the electrochemical process that occurs on a single wire is much more homogeneous than that involved in the entire WBE [18,19,20]. Tan groups used #70 steel to prepare WBE and demonstrated that the microelectrodes do not affect each other, and the corrosion potential and AC impedance values of each electrode could be measured independently and randomly [21]. Their group utilized WBE in monitoring, characterizing, and optimizing the polyaniline (PANI) electrodeposition process and evaluating the anticorrosion behaviors of the coatings [22]. Zhang et al. applied WBE to investigate the cathodic protection of defective areas of low-content (40 wt.%) zinc-rich coatings under immersion and atmospheric thin liquid film conditions [23]. 

In addition, WBE is also an accurate technique for investigating the effect of microorganisms on the corrosion behavior of metal substrates, especially for the speculation investigation of localized corrosion mechanisms under biofilms [24]. For example, Wang et al. developed a new device based on WBE for the first time to study the relationship between local potential/current patterns and H_2_O_2_ distribution in an artificial biofilm/stainless steel interface [25]. The results illustrated that the electrochemical homogeneity of biofilm-related systems is closely related to biofilm heterogeneity. Li et al. assisted the WBE technique to investigate the distribution of sulfate-reducing bacteria (SRB) inside the crevice and the effect of the crevice width on the SRB-induced corrosion of stainless steel 316 L. They demonstrated that the crevice width plays an essential role in corrosion metals in crevice environments with SRB, and the most severe corrosion occurred at the crevice that was 150 μm in width [26]. Because of the complex distribution of ionic currents in electrolytes during microbiologically induced corrosion (MIC), it is difficult to accurately detect the existence of heterogeneous biofilms covering the metal by traditional electrochemical techniques. As a practical tool for investigating the deterioration process of defective coatings and the corrosion process of steel substrates with microorganisms, WBE is one of the best choices because of its high accuracy and sensitivity features. 

In our previous work, the biodegradation effect of *Talaromyces funiculosus* (*T. funiculosus*), which was isolated from the atmospheric environment of the rainforest in Xishuang-banna (southern China), was studied in humid and hot atmospheric environments. Because of the organic acid secreted by *T. funiculosus* and its mycelium attacking, the PU coatings exhibited local damage and contributed to the presence of wrinkles, blisters, and even micro-holes [27]. However, we did not distinguish the barrier properties of these different areas. Herein, WBE and electrochemical impedance spectroscopy (EIS) tests were conducted to study the local degradation effect of *T. funiculosus* on PU coatings and the corrosion process of the metal substrates. The macro morphology of the coatings on each electrode after immersion with and without the *T. funiculosus* medium was observed by a handheld digital microscope. The corrosion morphology of some typical electrodes with and without being attacked by fungi was evaluated by scanning electron microscopy (SEM). The corrosion products under the broken coatings area were evaluated by Raman spectra. Using the WBE technique combined with the traditional valuation method for coatings, the differences in the biodegradation behavior in different regions of the PU coatings and the local corrosion behavior of metal substrates under the coatings can be accurately established. 

## 2. Materials and Methods

### 2.1. WBE Fabrication

The WBE was fabricated by Q235 carbon steel (diameter: 1 mm, interval: 0.8 mm) into a 10 × 10 matrix and sealed with epoxy resin, as shown in Figure 1. The surface area of each Q235 wire was about 0.785 mm^2^, and the entire working surface area of the WBE specimen was about 7.85 mm^2^. The WBE was abraded to 180 and 240 grit with silicon carbide papers, followed by cleaning with acetone and ethanol. The PU coating (Benchen, Shanghai, China) was applied on the working surface and then placed under a dust shield for dying. Each wire acted as both a mini-electrode (sensor) and as a substrate for corrosion. The terminal of 100 wires was connected to the computer cables to measure the current distributions on the WBE surface. The composition of the Q235 steel is Fe (98.68%), C (0.11%), Mn (0.35%), Si (0.14%), P (0.02%), and S (0.04%). The thickness of the coatings was ~150 µm.

### 2.2. T. funiculosus Spore Suspension

*T. funiculosus* was separated from an atmospheric test station in Xishuangbanna. The polymerase chain reaction (PCR)-amplified products of purified colonies were sent to Beijing RuiBiotech Co., LTD for rDNA-ITS sequencing, and the sequence alignment was performed with BLAST of the National Center of Biotechnology Information (NCBI). Ambiguously and incorrectly aligned positions were adjusted manually using the Chromas sequence editor. The MAFFT was used to perform multiple sequence alignment between ITS1–5.8S-IT2S rDNA sequences and representative sequences (GenBank, No. KU 994887; 99% sequence coverage; 100% similarity).

The surface of potato dextrose agar (PDA, potato 3 g/L, dextrose 20 g/L, agar 14 g/L) containing *T. funiculosus* mycelium was swiped with a sterile inoculation hook and put into 1 mL of potato dextrose broth (PDB; potato 5 g/L, dextrose 15 g/L, peptone 10 g/L, NaCl 5 g/L; the broth was autoclaved at 121 °C for 15 min) for incubation. The hyphae were filtered through four layers of sterile filter paper to obtain a spore suspension. The spore concentration of *T. funiculosus* was about ~10^6^ CFU/mL, as determined by optical microscopy (AxioLab.A1, Zeiss, Jena, Germany). 

### 2.3. Current Density Mapping

The spore suspension of *T. funiculosus* was diluted tenfold, and the PU-coated WBE was placed in this fungal liquid medium. The device was sealed with hot melt adhesion to prevent infection from other organisms, as shown in Figure 2. The catheter covered by sealing film was inserted into a rubber stopper to provide respiration for *T. funiculosus.* The three-electrode system consisted of WBE as the working electrode, a platinum foil as the counter electrode, and a saturated calomel electrode (SCE) as the reference electrode. The current density distribution was acquired after being applied by a wire beam electrode corrosion tester (CST 520, Jiangsu, China). 

### 2.4. EIS Measurements

The EIS was conducted using an electrochemical station (PARSTAT 2273, America) with a three-electrode system in sterilized and *T. funiculosus*-contained liquid mediums. A one-piece electrode was used as a working electrode, a platinum foil was the counter electrode, and SCE was the reference electrode. After the open-circuit potential (OCP) was stable, the EIS test was performed in the liquid medium at room temperature in a frequency range from 10^5^ to 10^−2^ Hz and with a sinusoidal perturbation of 20 mV vs. the OCP. All measurements were performed at least twice to guarantee repeatability. The impedance data were fit by the Zsimpwin software program.

### 2.5. Corrosion Morphology Characterization

After incubation in a sterilized and fungal liquid medium at 30 °C and 95% relative humidity (RH) for 14 days, the morphology of the coating on WBE was acquired by a handheld digital microscope (Dino-lite, Taiwan, China) and SEM (Zeiss Gemini 500, Jena, Germany). After removing the PU coatings, the morphology and composition of the corrosion products on the metal surface were evaluated by SEM (JEOL, JCM6000-PLUS, Tokyo, Japan), EDS, and Raman spectroscopy (LabRAM HR Evolution, HORIBA Jobin Yvon, Tokyo, Japan). The wavelength of the argon ion CW laser is 532 nm at room temperature, and the collection time is 60 s.

## 3. Results and Discussion

### 3.1. Current Density Distribution

#### 3.1.1. Sterile Environmental Analysis

After immersion in a sterile liquid medium for 14 days, the current density of the entire coated WBE surface was kept at a low level of around 10^−9^~10^−7^ A/cm^−2^, as shown in Figure 3. Specifically, after 2 days of immersion, the profiles were all uniform in the range of ~10^−8^ A/cm^−2^ (Figure 3b). As the immersion time increased, the current density increased slightly in some local regions (weak areas) of the WBE due to the electrochemical inhomogeneity of the coatings [28,29]. In Figure 3c, it was noteworthy that, among these weak areas, an obvious current peak was first found at the #74 (4, 8) electrode area, with about 1.09 × 10^−7^ A/cm^−2^ after 5 days of immersion. The profile increased to 2.10 × 10^−7^ A/cm^−2^ after immersion for 11 days, as shown in Figure 3e, which indicated that the corrosion would be initiated in this area. The current density of the #53 (3, 6) electrode area was about 2.48 × 10^−7^ A/cm^−2^, which would be developed into another potential localized corrosion site. Noticeably, although these two sites manifested as the weaker areas on the entire WBE surface, the current density was maintained at ~10^−7^ A/cm^−2^, indicating a good barrier effect of the coating. The current densities of other electrodes were kept at 10^−9^ A/cm^−2^ or even less, like the performance of the #40 (10, 4) electrode, during the 14 days of immersion. Hence, the above results indicated that the PU coating was intact and kept an outstanding shielding performance after the immersion in sterile liquid mediums for 2 weeks.

#### 3.1.2. Fungal Environmental Analysis

Figure 4 shows the current density of the entire WBE surface after the immersion in the *T. funiculosus*-contained liquid medium for the same time. Compared with Figure 3, the profiles were much higher than those of the sterile counterparts, with about 10^−6^~10^−3^ A/cm^2^ during the 14 days of immersion, except for the initial state. According to the current density variation, it can be divided into three stages. As for the first stage (0~2 days), there were four electrodes exhibiting an obvious lift in the current density, which were #62 (2, 7), #63 (3, 7), #91 (1, 10), and #94 (4, 10), with 2.96 × 10^−3^ A/cm^2^, and 2.95 × 10^−3^ A/cm^2^, 3.18 × 10^−3^ A/cm^2^, 3.15 × 10^−3^ A/cm^2^, respectively (Figure 4b). Combined with the number of weak electrodes after 5 days of immersion, the failure local area enlarged gradually, with the immersion time increasing during the second stage. Eight anode electrodes, including #57 (7, 6), #60 (10, 6), #67 (7, 7), #74 (4, 8), #76 (6, 8), #79 (9, 8), #85 (5, 9), and #92 (2,10), presented current densities of about 10^−3^ A/cm^2^ (Figure 4d) after the immersion for 8 days. By this time, the unattacked area acted as a large cathode, forming a corrosion cell with the anode electrode and accelerating the corrosion rate of these weak areas. In the third stage, the number of weak areas decreased to four, which were the #67, #74, #85 and #92 electrodes after immersion for 11 days, as shown in Figure 4e. This could be attributed to an accumulation of corrosion products under the coatings, which prevents the corrosive media from contacting the metal substrates and plays a corrosion inhibition role (Section 3.3) [29,30,31]. As for the other electrodes, the current density was about 10^−5^ A/cm^2^, like the #18 (8, 2) electrode during the whole 14 days of the immersion time. Compared with the profiles in Figure 3, the existence of *T. funiculosus* in a liquid medium apparently reduced the overall shielding performance of the PU coatings and induced localized corrosion on the WBE surface. 

### 3.2. Electrochemical Behavior of Selected Electrodes

#### 3.2.1. Changes in the Current Density

To further investigate the variation principle of the current density for the weak areas on the entire WBE surface, the #85 and #18 electrodes after the immersion in the *T. funiculosus*-contained medium and the #74 and #40 electrodes after the immersion in the sterile liquid medium were selected as the representatives. In Figure 5a, the #74 and #40 electrodes constantly exhibited a low current density during the entire period. In regard to the profiles of electrodes immersion in the *T. funiculosus*-contained medium, the current density of #18 was similar to that of the sterile counterparts during the 14 days of the immersion time, while the profiles of the #85 electrode fluctuated at ~10^−3^ A/cm^2^ from 5 days to 11 days, which was much higher than that of another electrode. This was convictive proof that the damage of the coatings was initiated locally. In detail, in Figure 5b, the current density increased from about 2.92 × 10^−3^ A/cm^2^ /cm^2^ for 5 days to about 3.14 × 10^−3^ A/cm^2^ for 11 days. The current density of the #18 electrode changed from about −1.30 × 10^−5^ A/cm^2^ to −7.88 × 10^−6^ A/cm^2^ after 11 days of immersion, in which the positive and negative profiles represented the anode and cathode area regarding the working electrode [32,33,34]. Noticeably, as the current density of the #85 electrode increased positively, the profiles of the #18 electrode decreased negatively, indicating that the surface of the #85 electrode served as an anode and that of #18 served as a cathode. Because the anode area was much smaller than the cathode area over the entire WBE surface, the current density of #85 was sharply increased at the beginning. With the time prolonging from the fifth day to the eighth day, the number of anode electrodes increased from four to eight, and the current density on #85 showed a slight downward trend, which indicated that the number of anode electrodes has a direct influence on the current density distributed on each electrode. Comparing the current density on the fifth day with that on the eleventh day, the current density of the electrode increased. It was determined that the corrosion process of the electrodes was intensified during the 11 days of immersion on the basis of the same number of anode electrodes for both periods. It was noticeable that the #85 electrode was transformed from local anodes to cathodes, and the current density decreased dramatically from 11 days to 14 days of immersion, which can be attributed to the accumulation of corrosion products, as presented in Figure 4e,f.

#### 3.2.2. EIS Analysis

The #74 electrode was selected as a representative of the coating immersed in a sterile liquid medium, and its EIS result is shown in Figure 6. It was found that the |Z|_0.01Hz_ value of the #74 electrode was maintained at ~10^10^ Ω·cm^2^ during the whole period, and only one time constant was observed in the corresponding phase angles plot (Figure 6b). With the increased immersion time, the line-point frequency shifted to a higher value, which revealed that the barrier properties of the coating were weakened slightly. Correspondingly, the semi-circles shown in the Nyquist plot changed a little as the immersion time increased (Figure 6c). The results above indicate that the sterile liquid medium has almost no effect on the shielding performance of the PU coating [35], which still exhibited outstanding corrosion protection after 14 days of immersion. The electrochemical equivalent circuit modeling shown in Figure 7a was utilized to fit the EIS results. Rs and Rc display the solution resistance and coating resistance, and the constancy phase elements Qc indicate the coating capacitance. During the whole 14 days of immersion, the Qc kept at a high level (~10^−10^ Ω^−1^·cm^2^·S^n^), and Rc was maintained at about ~10^10^ Ω·cm^2^, as shown in Table 1, which also indicated the intact barrier performance of the coatings [36]. This result agrees with the results exhibited in Figure 3. 

The EIS results for the #85 and #18 electrodes after the immersion in the *T. funiculosus*-contained medium for different periods are compared in Figure 8. In Figure 8a, the |Z|_0.01Hz_ value decreased to 1.06 × 10^9^ Ω·cm^2^ after 14 days of immersion, which was reduced by an order of magnitude compared with the sterile environment counterparts. Furthermore, during the two-week immersion period, a new time constant typically appeared in the low-frequency region (Figure 8b), and two semi-circles could be witnessed in the corresponding Nyquist plot (Figure 8c), reflecting that the liquid medium penetrates the coatings and that corrosion initiates on the metal substrates [37]. During this period, the electrochemical equivalent circuit modeling shown in Figure 7b was utilized to fit the EIS results, and the data are shown in Table 2. Rc decreased from 6.09 × 10^8^ Ω·cm^2^ to 5.33 × 10^8^ Ω·cm^2^, and Qc increased from 1.05 × 10^−10^ Ω^−1^·cm^2^·S^n^ to 1.26 × 10^−10^ Ω^−1^·cm^2^·S^n^. This indicated that the coating was permeated with liquid. Noticeably, during this process, the phase angle on the high-frequency region was still maintained at about 90°. This indicated that the macroscopic pores had not formed yet, but the bond between the coating and the metal substrate disappeared [33], which made a preferred condition for blistering, as the images illustrate in Figure 9 and Figure 10. Moreover, Rct, which represents the charge transfer resistance, increased from 5.67 × 10^8^ Ω·cm^2^ to 1.73 × 10^9^ Ω·cm^2^ from 3 days to 11 days of immersion. This revealed that the liquid penetrated into the coatings and reached the metal surface, contributing to the metal corrosion behaviors. The corrosion products accumulation under the coatings resulted in an increase in Rct. Compared with the change in the current densities, it was found that the |Z|_0.01Hz_ value showed a slight downward trend, with current density decreasing from 3.14 × 10^−3^ A/cm^2^ at 11 days to 1.54 × 10^−6^ A/cm^2^ at 14 days (Figure 5a). During this process, although the anode electrode may shift to a cathode one, as mentioned before, the barrier property of the coating on the #85 electrode cannot be recovered, which results in the unchanged value of |Z|_0.01Hz_. 

As for the #18 electrode that served as a cathode in the *T. funiculosus*-contained medium, its EIS results changed similarly to those of the #74 electrode obtained in the sterile liquid medium (Figure 6). The |Z|_0.01Hz_ values were about 5.12 × 10^10^ Ω·cm^2^, 4.92 × 10^10^ Ω·cm^2^, 1.30 × 10^10^ Ω·cm^2^, and 1.05 × 10^10^ Ω·cm^2^ after immersion for 3 days, 7 days, 11 days, and 14 days, respectively (Figure 8d). Combined with the profiles of the #85 electrode, the degradation of the PU coating caused by *T. funiculosus* was not uniform, so coatings on different electrodes showed different barrier performances. Moreover, the |Z|_0.01Hz_ value decreased by 3.87 × 10^10^ Ω·cm^2^ from immersion for 7 days to 14 days. Apparently, the profiles decreased less in the former 7 days than those presented in the latter 7 days, which may be because the self-corrosion process of the coating caused by *T. funiculosus* reduced its shielding effect during the later period. During this time, the self-corrosion caused by the failure of the coatings on this single electrode surface and the electric couple effect which was formed because of the uneven degradation and the different corrosion degree of the entire PU coating both existed. In the coupling system, the cathode areas could accelerate the corrosion behavior of the anode areas, and the number of anode electrodes decreased dramatically from eight to four in the second week. According to the results in Figure 4, the total current densities and anode areas decreased, which reflected that, for the failure of PU coatings, the corrosion behavior caused by the electric couple effect was reduced. However, in this period, the degradation effect of *T. funiculosus* on the coating cannot be ignored, which was the main reason for the corrosion acceleration. In Figure 8e, the phase plot was ~90°, and one time constant was witnessed, but the line-point frequency shifted to a higher value compared with the sterile counterpart (Figure 6b) with the increased immersion time. This revealed that the barrier properties of the coating weakened, and the reduced radius of the semi-circle arc also supported these results [37]. The fitting results shown in Table 3 also demonstrated this result. Specifically, Rc decreased from 6.25 × 10^10^ Ω·cm^2^ to 9.99 × 10^9^ Ω·cm^2^, and Qc increased from 1.21 × 10^−10^ Ω^−1^·cm^2^·S^n^ to 1.53 × 10^−10^ Ω^−1^·cm^2^·S^n^, indicating water infiltrating the coatings. However, compared with the profiles of the #85 electrode, the barrier property of the coating was much better. 

### 3.3. Morphology of Coatings

After 14 days of immersion in a sterile liquid medium, the morphologies of the entire coated WBE surface and some representative electrode surfaces were observed, as shown in Figure 9. Compared with the original state (Figure 9a), little change can be perceived on the entire surface of WBE after the immersion in the sterile liquid mediums (Figure 9b). More details of the 74# (4, 8) and 40# (10, 4) electrodes’ surfaces are enlarged in Figure 9c–f. In Figure 9c,e, no blisters or breakage were seen, indicating that the coatings on the #74 and #40 electrode surfaces were intact, which was also illustrated in the magnified morphology of the edge part (Figure 9d) and the center part (Figure 9f) of the electrodes.

Comparatively, after the immersion in a *T. funiculosus*-contained medium for 14 days, lots of blisters were observed on almost every electrode surface, as shown in Figure 10b. The #85 and #18 were selected as representative electrodes to observe more details of the coating surfaces. In Figure 10c (#85 electrode), varying-sized blisters spread almost over the entire surface. Moreover, in the enlarged images (Figure 10d), the brown corrosion products accumulated underneath the blister. This reflected that the PU coatings were delaminated from the metal substrate, the liquid mediums infiltrated into the coatings, and the metal surface was attacked. However, as for the #18 electrode sample, the number of blisters was less than that of those on the #85 electrode surface. Moreover, as mentioned the description in Section 3.1, the current density of the #85 electrode was much higher than that of the #18 electrode, which proved that the failure of PU coatings on the former was more severe than that of the latter. 

Comparing the morphology observed in the sterile mediums with that in the *T. funiculosus*-contained mediums, it could be determined that the existence of *T. funiculosus* accelerated the general corrosion process of PU coatings, and its inhomogeneous effect on the coating may exacerbate the localized damage of the coating as well. Theoretically, the coating damage in these local anode regions could be more severe from a microscopic perspective, especially for the #85 electrode. According to the previous work from our group, the ester bond hydrolysis that occurs in the PU structure can be accelerated because of the carboxylic acids secreted by *T. funiculosus*. Moreover, along with the hydrolysis of the ester and urethane bonds, carbon chains and ring structures were also broken by *T. funiculosus* [27]. 

Figure 11 shows more details of the surface morphologies of these electrodes after the immersion in different mediums for 14 days. In Figure 11a,b, the coatings on the #74 and #40 electrodes were uniform, without being damaged after the immersion in sterile mediums, which agreed with the handheld digital microscope images results (Figure 9). This demonstrated that the sterile liquid medium had little impact on the coating failures during the immersion, which agreed with the previous results. However, after the immersion in the fungal liquid medium, the morphologies of the #85 and #18 electrode surfaces were significantly different in the SEM images. In Figure 11c, the diameter of the blister was about 261 μm, and the center sank down, indicating that the existing *T. funiculosus* in the liquid mediums accelerated the peeling of the coatings. An irregular micro-crack located at the upper-right of the edge of the huge blister (Figure 11d) may act as a channel for the transfer of electrolytes to metal substrates and lead to the formation of this blister. This may be caused by the attack of mycelium and the coating degradation caused by secretions of *T. funiculosus* [27]. In Figure 11e, the coating on the #18 electrode surface was basically smooth, with some corrugated areas near the colonized individual spores of *T. funiculosus*. It can be found that the coating on the #18 electrode surface suffered less damage compared with the #85 counterpart. Finally, these areas were prone to the formation of blisters and corrosion products. Moreover, in comparison to these two electrodes, the failure of the coating was caused by localized damaged and peeling off after the impact on *T. funiculosus.*

### 3.4. Corrosion Production Evaluation

Figure 12 showed the morphology and corresponding EDS spectra of the corrosion products on the substrate after stripping the coating on the electrodes. According to Figure 12a,b, plenty of flake corrosion products were formed under the blisters after the immersion in the *T. funiculosus*-contained medium for 14 days. The EDS results showed that the corrosion products mainly consisted of Fe (26.0 wt.%), O (43.9 wt.%), and C (30.1 wt.%), indicating that γ-FeOOH may exist [38]. Furthermore, globular corrosion products consisting of Fe (14.2 wt.%), O (29.4 wt.%), and C (56.4 wt.%) were observed under the coating, as shown in Figure 12c,d. Referring to the previous literature, they could be α-FeOOH particles that nucleate and grow up based on the γ-FeOOH surface [39,40]. In addition, Cl^−^ coming from the PBD medium (5 g/L NaCl) may facilitate this process, and the detail was described in the following reactions:γ-FeOOH + Cl^−^ → FeOCl + OH^−^
FeOCl + H_2_O → α-FeOOH + HCl↑

Moreover, the high content of the C element on products indicated that the metabolite secreted by microorganisms was vestigial there [35]. The Raman spectra were performed to further determine the compositions of the corrosion products, and the results are shown in Figure 13. The peaks at 169 cm^−1^, 622 cm^−1^, and 1004 cm^−1^ belong to γ-FeOOH, γ-Fe_2_O_3_, and α-FeOOH, respectively [38]. These results basically agreed with the EDS and SEM results. 

## 4. Conclusions

In this work, the biodegradation effect of *T. funiculosus* on the corrosion failure of PU coating was investigated based on WBE in a simulated rainforest environment. The conclusions can be summarized as follows.
(1)The local current density distribution on the PU-coated WBE changed greatly after 14 days of immersion in the *T. funiculosus*-contained medium. After the immersion in sterile liquid mediums, the anode current density of the individual electrode was kept at around 10^−7^ A/cm^−2^, but it was only at about 10^−3^ A/cm^2^ for the *T. funiculosus*-contained counterpart.(2)The corrosion behaviors under the PU coatings after the immersion in a sterile medium were slight, with a |Z|_0.01Hz_ value at ~10^10^ Ω·cm^2^ (#74 electrode). However, the local corrosion behavior is obvious after the immersion in the *T. funiculosus*-contained medium, with |Z|_0.01Hz_ values at 1.05 × 10^10^ Ω·cm^2^ (#85 electrode) and 1.06 × 10^9^ Ω·cm^2^ (#18 electrode).(3)The corrosion behavior of metal substrates was made up of two parts, which were the self-corrosion process triggered by entries of corrosive media caused by the local damage of the coating and the corrosion process caused by the electric couple effect resulting from the uneven degradation of the coating.(4)The presented *T. funiculosus* accelerated the degradation of the PU coatings, causing micro-holes, cracks, and sinking on the coating surface, especially near the mycotic spore.(5)The existing *T. funiculosus* reduced the overall shielding performance of the PU coatings and contributed to corrosion behaviors concentrated in localized areas. The corrosion products consisted of γ-FeOOH, γ-Fe_2_O_3_, and α-FeOOH.

## Figures and Tables

**Figure 1 materials-16-01402-f001:**
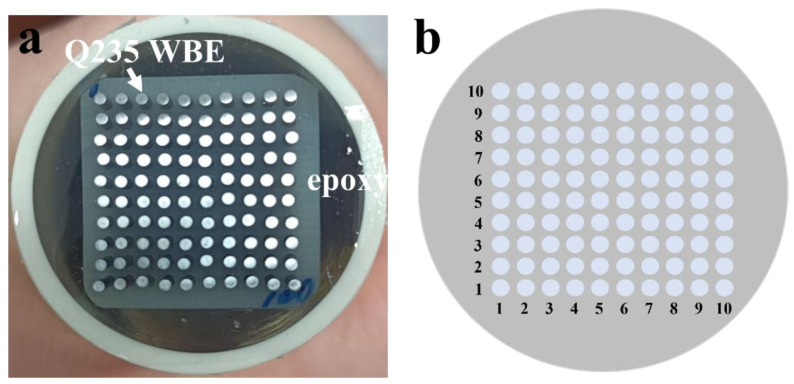
Photo of Q235 carbon steel WBE (**a**) and distribution of the WBE array (**b**).

**Figure 2 materials-16-01402-f002:**
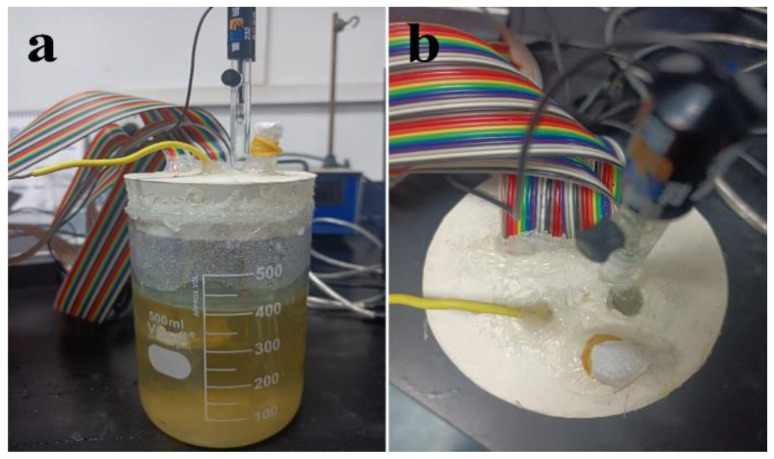
Photos of the front view (**a**) and top view (**b**) of the device for studying coating degradation and metal corrosion using WBE.

**Figure 3 materials-16-01402-f003:**
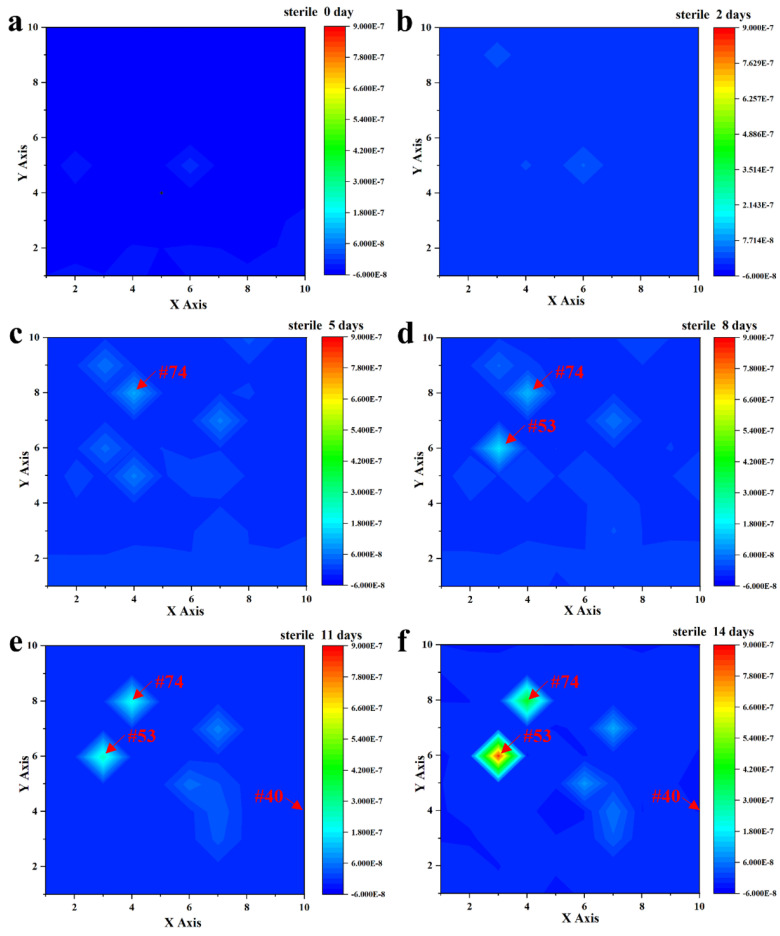
Current density distribution of the entire WBE surface after the immersion in a sterile liquid medium for 0 days (**a**), 2 days (**b**), 5 days (**c**), 8 days (**d**), 11 days (**e**), and 14 days (**f**).

**Figure 4 materials-16-01402-f004:**
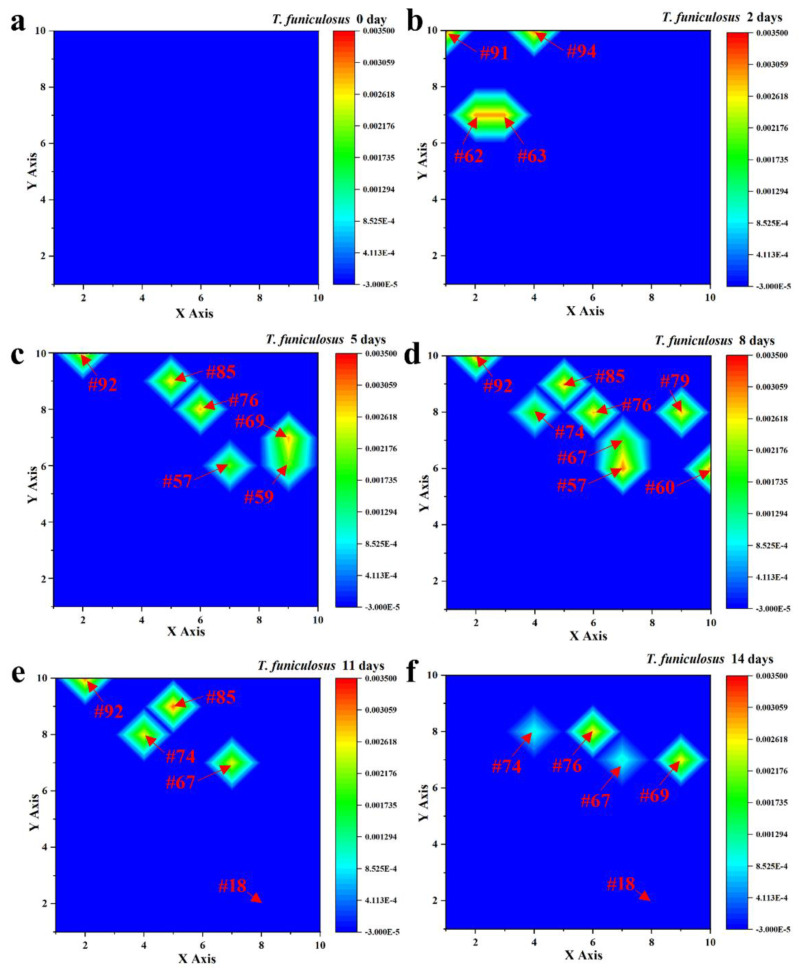
Current density distribution of the entire WBE surface after the immersion in the *T. funiculosus*-contained medium for 0 days (**a**), 2 days (**b**), 5 days (**c**), 8 days (**d**), 11 days (**e**), and 14 days (**f**).

**Figure 5 materials-16-01402-f005:**
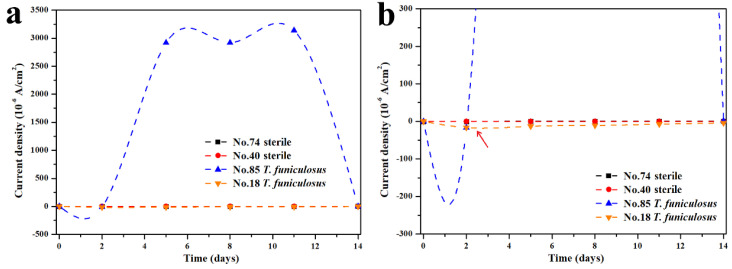
Variation in the current density of the #74 and #40 electrodes after the immersion in a sterile liquid medium and of the #85 and #18 electrodes after the immersion in the *T. funiculosus*-contained medium for 14 days (**a**) and the correspondingly partially enlarged line chart (**b**).

**Figure 6 materials-16-01402-f006:**
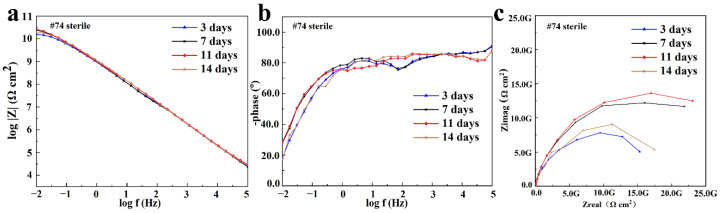
The Bode plot (**a**), phase plot (**b**), and Nyquist plot (**c**) of the #74 electrode after the immersion in a sterile liquid medium for 14 days.

**Figure 7 materials-16-01402-f007:**
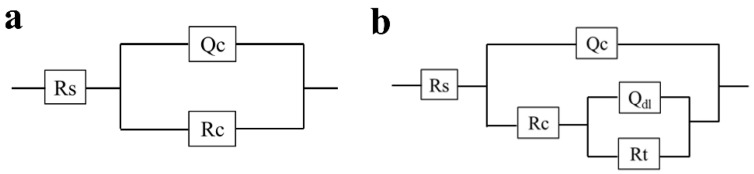
The equivalent circuit used to fit the EIS curves for the #74 electrode immersion in the sterile liquid medium and the #18 electrode immersion in the *T. funiculosus*-contained medium (**a**) and the #85 electrode immersion in the *T. funiculosus*-contained medium (**b**).

**Figure 8 materials-16-01402-f008:**
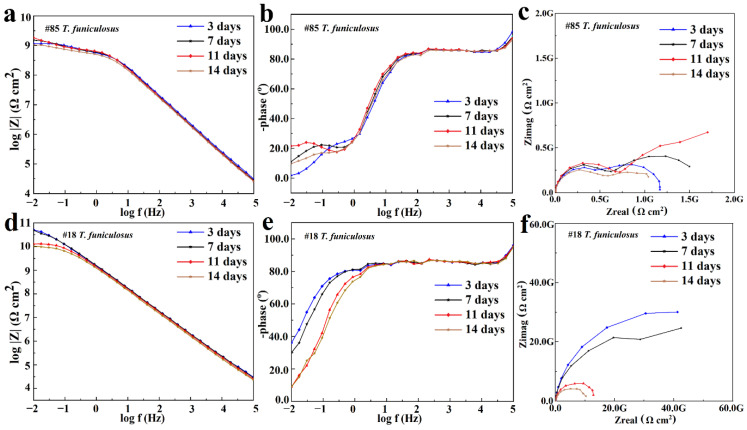
The Bode plot (**a**,**d**), phase plot (**b**,**e**), and Nyquist plot (**c**,**f**) of the #85 and #18 electrodes after the immersion in the *T. funiculosus*-contained medium.

**Figure 9 materials-16-01402-f009:**
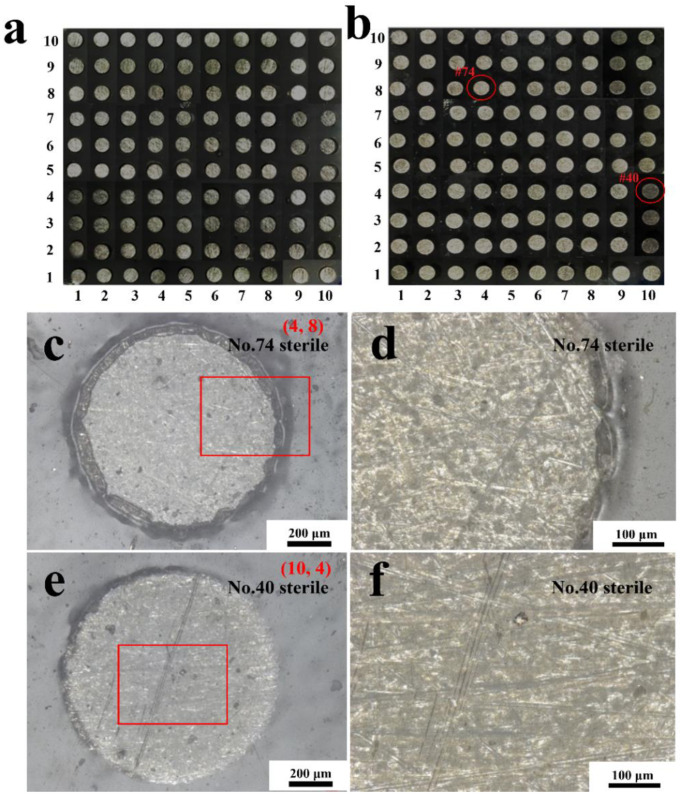
The handheld digital microscope images of the entire WBE surface at the initial state (**a**) and after the immersion in a sterile liquid medium for 14 days (**b**). The morphologies of the #74 electrode surface and its magnification (**c**,**d**) and #40 electrode surface and its magnification (**e**,**f**) after 14 days of immersion.

**Figure 10 materials-16-01402-f010:**
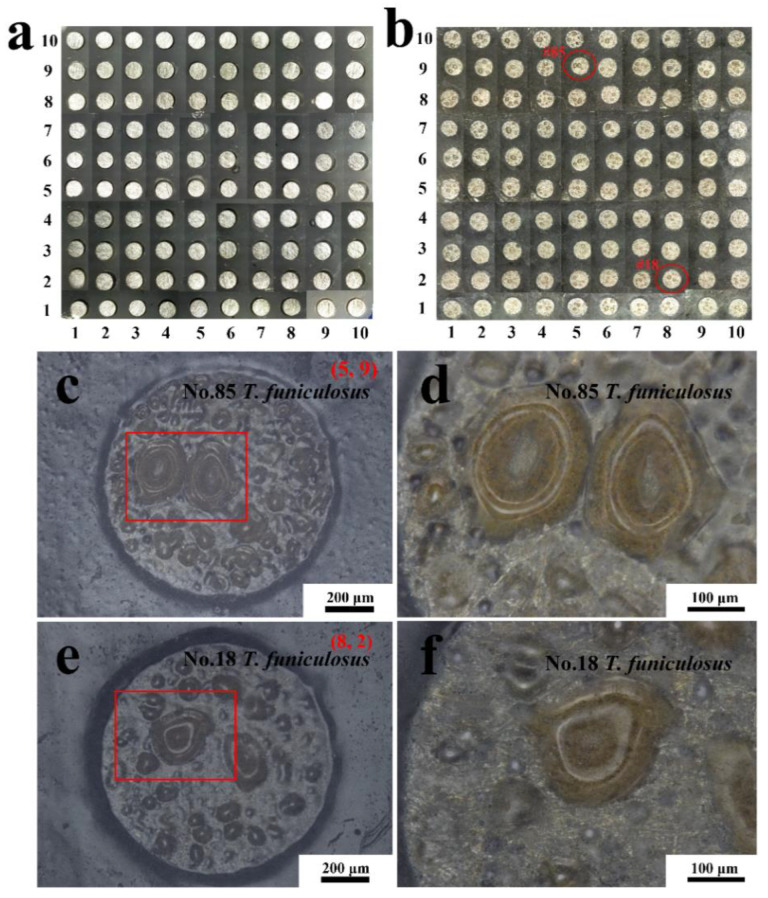
The handheld digital microscope images of the entire WBE surface at the initial state (**a**) and after the immersion in the *T. funiculosus*-contained medium for 14 days (**b**). The morphologies of the #85 electrode surface and its magnification (**c**,**d**) and #18 electrode surface and its magnification (**e**,**f**) after 14 days of immersion.

**Figure 11 materials-16-01402-f011:**
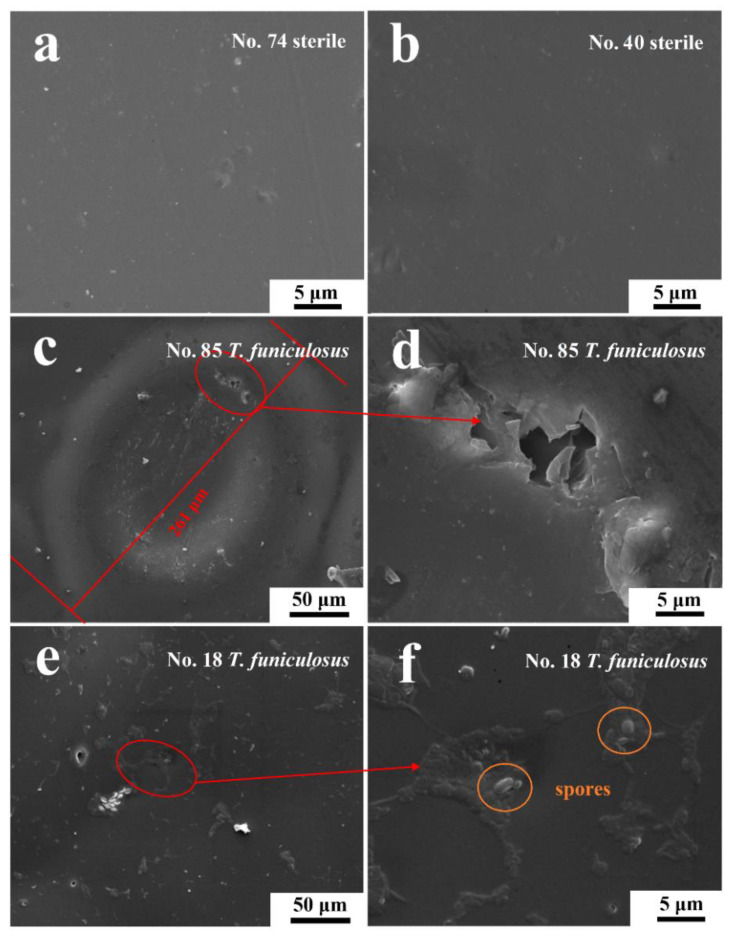
The SEM images of the #74 (**a**) and #40 electrode surfaces (**b**) after the immersion in sterile liquid mediums for 14 days and of the #85 (**c**,**d**) and #18 electrode surfaces (**e**,**f**) after the immersion in fungal liquid mediums for 14 days.

**Figure 12 materials-16-01402-f012:**
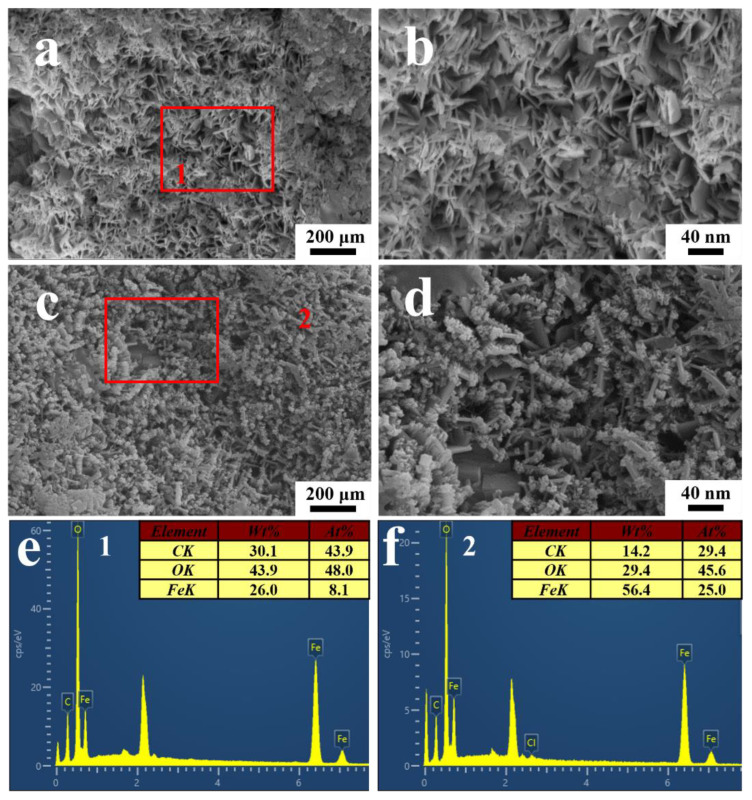
The SEM images of corrosion products underneath the PU coatings after the immersion in the *T. funiculosus*-contained medium for 14 days (**a**,**c**), and the corresponding enlarged images (**b**,**d**). The EDS spectra of the corrosion products mark on (**a**,**c**,**e**,**f**).

**Figure 13 materials-16-01402-f013:**
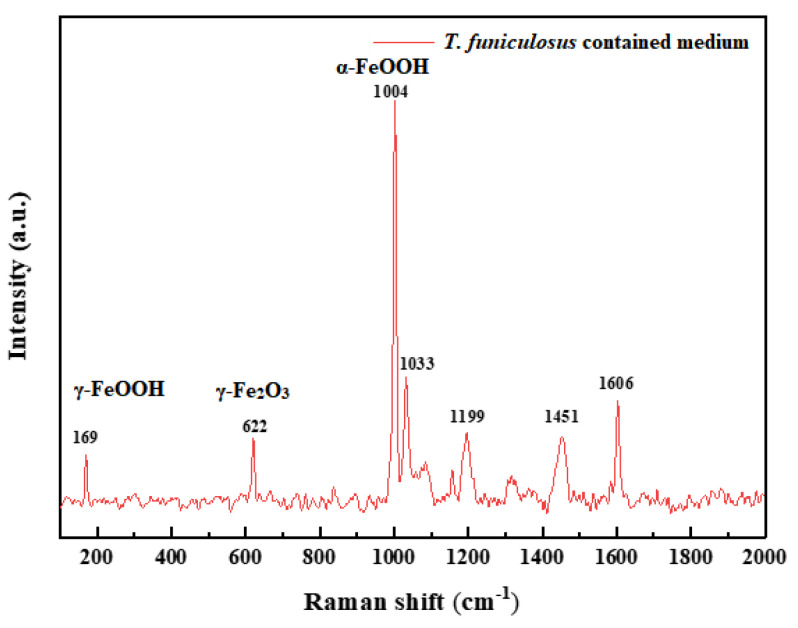
Raman spectra of corrosion products on the electrode after the immersion in the *T. funiculosus*-contained medium for 14 days.

**Table 1 materials-16-01402-t001:** EIS fitting results of the #74 electrode immersion in the sterile liquid medium for different times.

Immersion Time (d)	R_c_ (Ω·cm^2^)	CPE-Q_c_ (Ω^−1^·cm^2^·S^n^)	CPE-*n*
3	1.78 × 10^10^	2.33 × 10^−10^	0.895
7	2.88 × 10^10^	2.09 × 10^−10^	0.909
11	3.17 × 10^10^	2.11 × 10^−10^	0.896
14	1.99 × 10^10^	1.90 × 10^−10^	0.915

**Table 2 materials-16-01402-t002:** EIS fitting results of the #85 electrode immersion in the *T. funiculosus*-contained medium for different times.

Immersion Time (d)	R_c_ (Ω·cm^2^)	CPE-Q_c_ (Ω^−1^·cm^2^·S^n^)	CPE-*n*	R_ct_ (Ω·cm^2^)	CPE_2_-Q_dl_ (Ω^−1^·cm^2^·S^n^)	CPE_2_-*n*
3	6.09 × 10^8^	1.05 × 10^−10^	0.951	5.67 × 10^8^	1.68 × 10^−9^	0.939
7	6.53 × 10^8^	1.07 × 10^−10^	0.960	1.02 × 10^9^	3.08 × 10^−9^	0.820
11	7.05 × 10^8^	1.16 × 10^−10^	0.955	1.73 × 10^9^	3.96 × 10^−9^	0.751
14	5.33 × 10^8^	1.26 × 10^−10^	0.953	6.71 × 10^8^	3.91 × 10^−9^	0.707

**Table 3 materials-16-01402-t003:** EIS fitting results of the #18 electrode immersion in the *T. funiculosus*-contained medium for different times.

Immersion Time (d)	R_c_ (Ω·cm^2^)	CPE-Q_c_ (Ω^−1^·cm^2^·S^n^)	CPE-*n*
3	6.25 × 10^10^	1.21 × 10^−10^	0.945
7	4.71 × 10^10^	1.18 × 10^−10^	0.949
11	1.37 × 10^10^	1.31 × 10^−10^	0.950
14	9.99 × 10^9^	1.53 × 10^−10^	0.947

## Data Availability

No new data were created or analyzed in this study. Data sharing is not applicable to this article.

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
