# Peer review of "Investigating Different Local Polyurethane Coatings Degradation Effects and Corrosion Behaivors by Talaromyces funiculosus via Wire Beam Electrodes"

_materials, 2023, doi:10.3390/ma16041402_

Round 1
Reviewer 1 Report
The manuscript presents an interesting study about the degradation impact of T. funiculosus on the corrosion process of polyurethane coating by WBE. However, the paper needs major revisions before it is processed further, some comments follow:
Abstract:
The abstract must be improved. The abstract must contain information about the importance of the paper and the novelty of this study must be highlighted. Also, a short presentation of the methods used is needed.
Experimental section
Rename this section into Materials and methods
Please introduce the chemical composition of Q325 steel.
Introduce the software used for EIS studies.
The introduction section must be improved. In the last paragraph of the introduction section please highlight the novelty and the aim of this study, also add information about the characterization methods used and the main conclusion.
Results and Discussion
Please introduce the equivalent circuits corresponding to Nyquist and Bode plots and discuss them. Also, introduce the determined parameters into a table and discuss them, please see an example: DOI: 10.3390/app11177802
Also, figures 6 and 7 are not clear. Please replace them.
Conclusions
The conclusion looks vague. Add quantitative results and also limitations and suggestions.
Author Response
Abstract:
Comment 1: The abstract must be improved. The abstract must contain information about the importance of the paper and the novelty of this study must be highlighted. Also, a short presentation of the methods used is needed.
Response 1: According to the reviewer's suggestion, we have already modified the abstract carefully. In the abstract, we added the background and critical thinking in this manuscript. Moreover, the main results of this study have been modified in this part. The revised abstract is as follows.
The degradation effect of mold on the coating in a hot and humid environment is one of the important factors that cause layer failure. combined with the wire beam electrode (WBE) and the traditional surface analysis technique, the local biodegradation of the coatings and corrosion behaviors of metal substrates can be characterized accurately by a WBE. Herein, a WBE was used to study the degradation impact of Talaromyces funiculosus (T. funiculosus) isolated from a tropical rainforest environment on corrosion of polyurethane (PU) coating. After immersion for 14 days, the local current density distribution of the WBE surface can reach ~10-3 A/cm2 in the fungal liquid mediums but maintain ~10-7 A/cm2 in sterile liquid mediums. The |Z|0.01 Hz value of the high current densities area (#85 electrode) was 1.06× 109 Ω⋅cm2 in a fungal liquid medium after 14 days of immersion. After being attacked by T. funiculosus, the degradation of the PU was more severe and there were wrinkles, cracks, blisters, and even microholes distributed randomly on the surface of electrodes. This resulted based on self-corrosion caused by T. funiculosus degradation of the coating, the corrosion caused by the electric coupling effect of the coating was introduced. Energy dispersive spectroscopy (EDS), and Raman spectra results showed the corrosion products were flake and globular, which consisted of γ-FeOOH, γ-Fe2O3, and α-FeOOH.
Experimental section:
Comment 2: Rename this section into Materials and methods. Please introduce the chemical composition of Q325 steel.
Response 2: According to the reviewer's suggestion, we have revised the section 2 experimental section to Materials and method. moreover, we added the composition of Q235 steel in the 2.1 WBE fabrication section. The composition of the Q235 steel Fe (98.68%), C (0.11%), Mn (0.35%), Si (0.14%), P(0.02%), and S (0.04%).
Comment 3: Introduce the software used for EIS studies.
Response 3: The EIS was conducted by an electrochemical station (PARSTAT 2273, America) and the data were shown in PowerSuite and drawn by origin software. The software program Zsimpwin was applied to fit the impedance data. The information on the EIS studies was added in Section 2.4. EIS measurements
Comment 4: The introduction section must be improved. In the last paragraph of the introduction section please highlight the novelty and the aim of this study, also add information about the characterization methods used and the main conclusion.
Response 4: According to the reviewer's suggestion, we have modified the introduction, especially the last paragraph carefully. In this manuscript, we aim to share the degradation effect of T. funiculosus on PU coatings and the corrosion process of the metal substrates. Using the WBE technique combined with the traditional valuation method for coatings, the differences in biodegradation behavior in different regions of the PU coatings and the local corrosion behavior of metal substrates under the coatings can be accurately established. We have already heightened them in the last paragraph.
Results and Discussion
Comment 5: Please introduce the equivalent circuits corresponding to Nyquist and Bode plots and discuss them. Also, introduce the determined parameters into a table and discuss them, please see an example: DOI: 10.3390/app11177802
Response 5: Accrodin to the reviewer’s suggestion, we added the equivalent circuits in the mansucript. The eletrochemical equivalent circuit modeling shown in Figure 8a was utilized to fit the EIS reults for steirle liquid medium. Rs and Rc display the solution resistance and coating resistance, and constance phase elements Qc indicate coating capacitance. During the whole 14 days of immersion, the Qc kept at a high level (~10-10 Ω-1·cm2·Sn) and Rc mantainted about ~1010 Ω·cm2 as shown in table 1, which also indicated that the intact barrier peformance of cotings [35]. This result agreed with the results exhibited in Figure 3. As for #18 eletrode after immersion in T. funiculosus contained medium, the fitting reuslts shown in Table 3, which agreed with the results shown in Figure 8. Specificlly, the Rc decrased from 6.25×1010 Ω·cm2 to 9.99×109 Ω·cm2, and the Qc increased from 1.21×10-10 Ω-1·cm2·Sn to 1.53×10-10Ω-1·cm2·Sn, indicating water infiltrate into the coatings. As for 85# eletrode, the eletrochemical equivalent circuit modeling shown in Figure 7b was utilized to fit the EIS reults and data was shown in Table 2. During this time, Rc decreased from 6.09×108 Ω·cm2 to 5.33×108 Ω·cm2 and Qc increased from 1.05×10-10 Ω-1·cm2·Sn to 1.26×10-10 Ω-1·cm2·Sn. It indicated the coating was permeated with liquid. Morever, Rct which represented the charge transfer resistance increased from 5.67×108 Ω·cm2 to 1.73×109 Ω·cm2 from 3 days to 11 days of immersion. It revaled that liquid prentoren into coatings and reached to metal surface, contributing metal corrosion be-haviors. The corrosion products acculmulation under the coatings resulted in increased of Rct.
Figure 7. The equivalent circuit used to fit the EIS curves for #74 electrode immersion in sterile liquid medium and #18 eletrode immeresion in T. funiculosus contained medium(a), and #84 eletrode immersion in T. funiculosus contained medium (b).
Table 1. EIS fitting results of #74 eletrode immersion in sterile liquid medium for different times.
|
Immersion Time (d) |
Rc (Ω·cm2) |
CPE-Qc (Ω-1·cm2·Sn) |
CPE-n |
|
3 |
1.78×1010 |
2.33×10-10 |
0.895 |
|
7 |
2.88×1010 |
2.09×10-10 |
0.909 |
|
11 |
3.17×1010 |
2.11×10-10 |
0.896 |
|
14 |
1.99×1010 |
1.90×10-10 |
0.915 |
Table 2. EIS fitting results of #85 eletrode immersion in T. funiculosus contained medium for different times.
|
Immersion Time (d) |
Rc (Ω·cm2) |
CPE-Qc (Ω-1·cm2·Sn) |
CPE-n |
Rct (Ω·cm2) |
CPE2-Qdl (Ω-1·cm2·Sn) |
CPE2-n |
|
3 |
6.09×108 |
1.05×10-10 |
0.951 |
5.67×108 |
1.68×10-9 |
0.939 |
|
7 |
6.53×108 |
1.07×10-10 |
0.960 |
1.02×109 |
3.08×10-9 |
0.820 |
|
11 |
7.05×108 |
1.16×10-10 |
0.955 |
1.73×109 |
3.96×10-9 |
0.751 |
|
14 |
5.33×108 |
1.26×10-10 |
0.953 |
6.71×108 |
3.91×10-9 |
0.707 |
Table 3. EIS fitting results of #18 eletrode immersion in T. funiculosus contained for different times.
|
Immersion Time (d) |
Rc (Ω·cm2) |
CPE-Qc (Ω-1·cm2·Sn) |
CPE-n |
|
3 |
6.25×1010 |
1.21×10-10 |
0.945 |
|
7 |
4.71×1010 |
1.18×10-10 |
0.949 |
|
11 |
1.37×1010 |
1.31×10-10 |
0.950 |
|
14 |
9.99×109 |
1.53×10-10 |
0.947 |
Comment 6: Also, figures 6 and 7 are not clear. Please replace them.
Response 6: According to the reviewer's suggestion, we have revised Figure 6 and Figure 7 to make it more clear. The new Figures are as follows.
Figure 6. The Bode plot (a), phase plot (b), and Nyquist plot (c) of the #74 electrode after immersion in a sterile liquid medium for 14 days.
Figure 7. The Bode plot (a, b), phase plot (c, d), and Nyquist plot (e, f) of the #85 and #18 electrodes after immersion in T. funiculosus contained medium.
Conclusions
Comment 7: The conclusion looks vague. Add quantitative results and also limitations and suggestions.
Response 7: According to the reviewer's suggestion, we have modified the conclusion and added the quantitative results of EIS in this part. Because we didn’t have any results regarding the effects of the biodegradation of Talaromyces funiculosus on modified coatings, we cannot provide a specific design scheme and suggestions. However, in future research work, we could try to add two-dimensional material (such as graph oxide and Mxene ) and antibacterial agents (such as Ag nanoparticles) to improve the anti-biodegradation ability of the coating, so as to achieve the purpose of inhibiting metal corrosion.

Reviewer 2 Report
Excellent work in the field of degradation of PU coatings due to Talaromyces funiculosus. There are a few comments in the attached file please address them.
Is there any way to stop this failure?
Is figure 3 simulation result?

Author Response
Excellent work in the field of degradation of PU coatings due to Talaromyces funiculosus. There are a few comments in the attached file please address them.
Comment 1: Is there any way to stop this failure?
Response 1: Like the reviewers pointed out, there are many ways to inhibit coating failure according to the published research. According to the research results of this manuscript, agents that inhibit microbial agents could be helpful in inhibiting coating failure. However, because we didn’t have any results regarding the effects of the biodegradation of Talaromyces funiculosus on modified coatings, we cannot provide a specific design scheme. In future research work, we could try to add two-dimensional material (such as graph oxide and Mxene ) and antibacterial agents (such as Ag nanoparticles) to improve the anti-biodegradation ability of the coating, so as to achieve the purpose of inhibiting metal corrosion.
Comment 2: Is figure 3 simulation result?
Response 2: Actually, the results shown in Figure 3 were the experimental results. the current density distribution of the entire WBE surface was conducted by a wire beam electrode corrosion tester (CST 520, China). The data were analyzed by the origin software and illustrated in Figure 3.

Reviewer 3 Report
Comments on the paper (materials-2107637) with the title "Investigated degrading effect of Talaromyces funiculosus on corrosion process of polyurethane coatings by wire beam electrode ", by Xiangping Hao , Kexin Yang , Dawei Zhang and Lin Lu
In their manuscript, the authors investigate the influence of Talaromyces funiculosus on the corrosion of a steel (Q235 carbon steel) protected with a polyurethane coating. For this purpose, they used the interesting approach of wire beam electrode technology. Current density mapping and electrochemical impedance spectroscopy were used as electrochemical measurement methods. The morphology was investigated with light microscopy and electron microscopy. The corrosion products were studied by EDS and Raman spectroscopy. Although the manuscript contains interesting results, I cannot recommend it for publication in its current state. I even suggest to reject it. However, I see the potential to publish it after a fundamental revision, which, in my opinion, also requires new experiments.
My rejection is based on the following reasons: In this manuscript, if I understood it correctly, the influence of Talaromyces funiculosus on the polyurethane (PU) coating and thus on corrosion is to be investigated. In the introduction, even possible degradation processes of PU are described also with respect to biofilms. In whole manuscript these important things are not investigated at all. There is no characterization of PU coating for chemical changes. The thickness of the coating is not even given. Does the chosen coating method guarantee the production of a dense compact PU film?
After microscopic characterization, only localized damage, existing micro-cracks and delamination are mentioned. With the help of the electrochemical investigations, a degradation effect by the fungus was also concluded. However, the underlying processes due to the probable interaction with Talaromyces funiculosus are then not addressed at all. The question is at which sites does the fungus attack and how does the micro-cracks and blisters develop?
In my opinion, the coating must also be examined spectroscopically and microscopically bevor interaction with the fungus. After the reaction, the coating should also be examined spectroscopically and not only the corrosion products of the steel. Are there perhaps already micro-cracks before the interaction or other sites where the T. funiculosus can attack, and what do the changes in the chemical structure of the Pu look like? The authors investigated the corrosion products with Raman spectroscopy, which could also be used to characterize the PU coating. Then the influence of the coating thickness should also be investigated. Furthermore, in my opinion, it would still have to be worked out whether there are differences in the corrosion with and without coating. For example, are other reaction products observed?
After this rather general criticism, I now have another question and a suggestion.
The authors have stated that the fungus interacts rather inhomogeneously with the coating, which could be shown by the differences in current densities at the individual electrodes. Is there any way to determine the concentration of the fungus at these sites? Perhaps this could provide further insight into the interaction mechanism.
I would suggest characterizing the films, if possible, using confocal laser scanning microscopy. This would be a good way to show changes in the film thickness and insights into the blisters and cracks.
Author Response
Comments on the paper (materials-2107637) with the title "Investigated degrading effect of Talaromyces funiculosus on corrosion process of polyurethane coatings by wire beam electrode ", by Xiangping Hao , Kexin Yang , Dawei Zhang and Lin Lu
In their manuscript, the authors investigate the influence of Talaromyces funiculosus on the corrosion of a steel (Q235 carbon steel) protected with a polyurethane coating. For this purpose, they used the interesting approach of wire beam electrode technology. Current density mapping and electrochemical impedance spectroscopy were used as electrochemical measurement methods. The morphology was investigated with light microscopy and electron microscopy. The corrosion products were studied by EDS and Raman spectroscopy. Although the manuscript contains interesting results, I cannot recommend it for publication in its current state. I even suggest to reject it. However, I see the potential to publish it after a fundamental revision, which, in my opinion, also requires new experiments.
My rejection is based on the following reasons: In this manuscript, if I understood it correctly, the influence of Talaromyces funiculosus on the polyurethane (PU) coating and thus on corrosion is to be investigated. In the introduction, even possible degradation processes of PU are described also with respect to biofilms. In whole manuscript these important things are not investigated at all.
Comment 1: There is no characterization of PU coating for chemical changes. The thickness of the coating is not even given. Does the chosen coating method guarantee the production of a dense compact PU film?
Response 1: Like the reviewers' opinion, we should provide the characterization like chemicals and thickness of coatings. In this manuscript, we added the thickness of the coatings in section 2.1, and the thickness of the coatings was ~150 μm. Because we report a paper that investigated the impact of degradation on PU by Talaromyces funiculosus and Phanerochaete chrysosporium from their secretions and mycelial attacks, and we have already discussed the chemical characterization changes in that paper. Hence, we didn’t exhibit that part in this manuscript. However, like the reviewer's suggestion, we should provide some information about that to make readers understand this work easily. So, we cited recent reports to give readers a more detailed picture of the progress of the work [37]. The FITR spectra of the coatings was illustrated in Figure 5a and 5b. According to the results shown in that paper, the ester bond hydrolysis that occurs in the PU structure can be accelerated because of carboxylic acids secreted by T. funiculosus. Moreover, along with the hydrolysis of ester and urethane bonds, carbon chains and ring structures were also broken by T. funiculosus. We prepared the coating according to GB/T1765–89 and then placed it under a dust shield for drying. According to the current density (Figure 3), EIS (Figure 6) results, SEM images (Figure 10a and b), and handheld digital microscope images (Figure 8), in sterile conditions shown in the manuscript, it can be found that the PU coatings had outstanding barrier properties and difficulty of water penetration, which indicated their excellent density compactness.[37] X. Hao, K. Yang, D. Zhang, L. Lu, Insight into Degrading Effects of Two Fungi on Polyurethane Coating Failure in a Simulated Atmospheric Environment. Polymers, 2023, 15, 328. https://doi.org/10.3390/polym15020328
Comment 2: After microscopic characterization, only localized damage, existing micro-cracks and delamination are mentioned. With the help of the electrochemical investigations, a degradation effect by the fungus was also concluded. However, the underlying processes due to the probable interaction with Talaromyces funiculosus are then not addressed at all. The question is at which sites does the fungus attack and how does the micro-cracks and blisters develop?
Response 2: As the reviewers said, the probable interaction with T. funiculosus and PU coatings was important, but we had already investigated that in our previous work [37]. That work demonstrated that the carboxylic acids secreted by T. funiculosus accelerated the hydrolysis of the ester and urethane bonds and damaged the carbon chain and benzene rings of the PU coating skeleton. Moreover, the mycelium penetrated the interface through microholes during the colonization of T. funiculosus. Hence, the combination of secretion of T. funiculosus and mycelium attack led to the failure of the coating and generated blisters. However, in our previous work, although we focused on the failure mechanism of the coating by T. funiculosus, we emphasized the results from the entire PU coatings instead of the local changes of the PU coatings. In this work, we would like to investigate the microscopic characterization from the view of localized damage and barrier properties of the PU coatings. After considering the reviewer’s suggestion, we thought it would be more helpful to include the results of the previous work in Figure 10. Hence, we cited some important conclusions from our previous work in this manuscript.
[37] X. Hao, K. Yang, D. Zhang, L. Lu, Insight into Degrading Effects of Two Fungi on Polyurethane Coating Failure in a Simulated Atmospheric Environment. Polymers, 2023, 15, 328. https://doi.org/10.3390/polym15020328
Comment 3: In my opinion, the coating must also be examined spectroscopically and microscopically bevor interaction with the fungus. After the reaction, the coating should also be examined spectroscopically and not only the corrosion products of the steel. Are there perhaps already micro-cracks before the interaction or other sites where the T. funiculosus can attack, and what do the changes in the chemical structure of the Pu look like? The authors investigated the corrosion products with Raman spectroscopy, which could also be used to characterize the PU coating. Then the influence of the coating thickness should also be investigated. Furthermore, in my opinion, it would still have to be worked out whether there are differences in the corrosion with and without coating. For example, are other reaction products observed?
Response 3: These parts include the surface of PU coatings and their chemical structure changes before and after colonization by T. funiculosus investigated in our previous work. The FTIR spectra were conducted, and we had already explained in response 2. That is the carboxylic acids secreted by T. funiculosus accelerated the hydrolysis of the ester and urethane bonds and damaged the carbon chain and benzene rings of the PU coating skeleton. The initial PU coatings are intact and there were no micro-cracks before T. funiculosus colonization, which was evaluated by SEM in our previous work (Figure S3) [37]. In regards to the influence of the coating thickness factors, we will furtherly investigate this in our future work. Because this work was focused on local degradation and local corrosion behaviors, we are afraid if a lot of text was spent discussing the effect of coating thickness, the main idea of the whole article will be unclear. Hence, although the thickness of the coatings may influence the degradation of the coatings, we didn’t discuss it in this manuscript. As for the corrosion products on bare substrates in fungal environments, we thought this part is not the main part of this work. Herein, we focused on the local biodegradation of the coatings and corrosion behaviors of metal substrates instead of the differences in the substrates' corrosion behaviors with and without PU protection. Hence, we didn’t discuss the corrosion product on bare metal substrates after colonization by T. funiculosus.
[37] X. Hao, K. Yang, D. Zhang, L. Lu, Insight into Degrading Effects of Two Fungi on Polyurethane Coating Failure in a Simulated Atmospheric Environment. Polymers, 2023, 15, 328. https://doi.org/10.3390/polym15020328
Comment 4: After this rather general criticism, I now have another question and a suggestion. The authors have stated that the fungus interacts rather inhomogeneously with the coating, which could be shown by the differences in current densities at the individual electrodes. Is there any way to determine the concentration of the fungus at these sites? Perhaps this could provide further insight into the interaction mechanism. I would suggest characterizing the films, if possible, using confocal laser scanning microscopy. This would be a good way to show changes in the film thickness and insights into the blisters and cracks.
Response 4: As the reviewers said, the local concentration of T. funiculosus is impact the local degradation of PU coatings, and contributed to the difference of current densities at individual electrodes. However, because the distribution of T. funiculosus on PU coating is random and we used continuous monitoring to obtain electrochemical data. During this process, the electrodes were immersed in closed fungal liquid medium devices, and we cannot obtain the concentration of the fungal at the corresponding location. Nevertheless, we will seriously consider the reviewer's suggestions and try to solve this problem in our future work. In this work, we have already tried to utilize confocal laser scanning microscopy to investigate the characterizing the films (as follows), but because these areas where T. funiculosus colonization is different from the WBE surface, it is difficult to build a relationship with biofilms or concentration of the T. funiculosus and the blisters and cracks. However, in our previous work, we had already verified that the impact of fungi on the coating failure is proportional to their concentration, especially when PU is the only available carbon resource to support fungi metabolism. Moreover, better-developed mycelia could secret more organic acids to accelerate the biodegradation of PU coatings, resulting in the coating not being able to resist the mycelium attack and leading to perforations of the coating. The surface of the PU coating was covered with a network of mycelium, the spores of T. funiculosus were gray-green, and the mycelium is white. With the extension of incubation time, the number of mycelia containing corrosion products increased, and the color of rust deepened. The thick mycelium as shown in Figure R1e is more yellow, which means thick mycelium is relatively more powerful and penetrable, and more aggressive to the coating.

Round 2
Reviewer 1 Report
The manuscript was improved accordingly to my comments. The manuscript can be published in the present form.
Author Response
Comment: The manuscript was improved accordingly to my comments. The manuscript can be published in the present form.
Response: Thanks for the reviewer effort on this manuscript.

Reviewer 3 Report
Comments on the revised paper (materials-2107637) with the title "Investigated degrading effect of Talaromyces funiculosus on corrosion process of polyurethane coatings by wire beam electrode ", by Xiangping Hao , Kexin Yang , Dawei Zhang and Lin Lu
At the beginning I would like to thank for the detailed answers of the authors. However, I cannot hide my disappointment that in the original manuscript the preliminary work (now citation [37]) was not mentioned with a word. There one can already ask the question, what is the really new information in this paper, just because the title of the two papers shows clear similarities. Furthermore, both conclusions also start similarly: "In this work, the biodegradation effect of T. funiculosus on the corrosion failure of PU 433 coatings was investigated" to [37] "In the present work, the biodegradation effects of two types of fungi (T. funiculosus
and P. chrysosporium) on the corrosion failure of the PU coatings were investigated in a simulated atmospheric environment." I, as a reader, think with the current heading that the effect of the fungus on the coating is examined. However, this effect was already partly described in [37]: "Thus, it was established that the colonization of T. funiculosus on the coating surfaces could destroy the integrity of the coating and cause local damage, leading to a reduction in corrosion protection by the coating". In my opinion, the authors investigated here the effects of the wrinkles, blisters and holes caused by the fungus on the local corrosion. To what extent the fungus still plays a role cannot yet be said, since the fungus, as the authors write, is randomly distributed over the surface and cannot be correlated with the randomly distributed holes at that time. In addition, the effects on the underlying steel were investigated more here. Perhaps the authors should therefore come up with a new title. In addition, the previous paper [37] should already be mentioned in the introduction as a preliminary work. This should make clear which new direction this manuscript has compared to it. For example, one could point out that one has already studied the degradative effect of the fungus on the polyurethane and now wants to look more closely at the local effects of the holes formed on the corrosion with the WBE electrode.
Further steps must also be explained before publication. In [37], EIS measurements have also been performed on the same material combination. What leads to clearly different IZI 0.01Hz for the global measurements in [37]. How do the authors explain the differences? Does it have something to do with the local measurement?
Then the caption for figure 8 needs to be changed. Here, for example, (a,d) are the Bode plots and not (a,b).
Author Response
Reviewer 3#:
Comments on the revised paper (materials-2107637) with the title "Investigated degrading effect of Talaromyces funiculosus on corrosion process of polyurethane coatings by wire beam electrode ", by Xiangping Hao , Kexin Yang , Dawei Zhang and Lin Lu
Comment 1:
At the beginning I would like to thank for the detailed answers of the authors. However, I cannot hide my disappointment that in the original manuscript the preliminary work (now citation [37]) was not mentioned with a word. There one can already ask the question, what is the really new information in this paper, just because the title of the two papers shows clear similarities. Furthermore, both conclusions also start similarly: "In this work, the biodegradation effect of T. funiculosus on the corrosion failure of PU 433 coatings was investigated" to [37] "In the present work, the biodegradation effects of two types of fungi (T. funiculosus and P. chrysosporium) on the corrosion failure of the PU coatings were investigated in a simulated atmospheric environment." I, as a reader, think with the current heading that the effect of the fungus on the coating is examined. However, this effect was already partly described in [37]: "Thus, it was established that the colonization of T. funiculosus on the coating surfaces could destroy the integrity of the coating and cause local damage, leading to a reduction in corrosion protection by the coating". In my opinion, the authors investigated here the effects of the wrinkles, blisters and holes caused by the fungus on the local corrosion. To what extent the fungus still plays a role cannot yet be said, since the fungus, as the authors write, is randomly distributed over the surface and cannot be correlated with the randomly distributed holes at that time. In addition, the effects on the underlying steel were investigated more here. Perhaps the authors should therefore come up with a new title. In addition, the previous paper [37] should already be mentioned in the introduction as a preliminary work. This should make clear which new direction this manuscript has compared to it. For example, one could point out that one has already studied the degradative effect of the fungus on the polyurethane and now wants to look more closely at the local effects of the holes formed on the corrosion with the WBE electrode.
Response 1: Like the reviewers’ opinion, we didn’t exhibit the continuity of the relevant work obviously and may contribute to the encounter obstacles in reading process for readers. According to the reviewer’s suggestion, we revise the title and the last paragraph of introduction in this manuscript carefully to solve this problem. The title revised to “Investigate different local polyurethane coatings degradation effect and corrosion behaviors by Talaromyces funiculosus via wire beam electrode”, to point out the local degradation effect of the PU coatings and its corrosion process of metal substrates under these areas.
Furthermore, we also added the explanation in the last paragraph of introduction to tell the different between this manuscript and the lasted report (reference [37] which was changed to [27]). The added parts were as follows. “In our previous work, the biodegradation effect of Talaromyces funiculosus (T. funiculosus) which was isolated from the atmospheric environment of the rainforest in Xishuang-banna (southern China) was studied in humid and hot atmospheric environ-ments. Because of the organic acid secreted by T. funiculosus and its mycelium attacking, the PU coatings exhibited local damage and contributed to the presents of wrinkles, blisters, and even micro-holes [27]. However, we didn’t distinguish the barrier properties of these different areas. Herein, the WBE and electrochemical impedance spectroscopy (EIS) tests were conducted to study the local degradation effect of T. funiculosus on PU coatings and the corrosion process of the metal substrates.”
Comment 2: Further steps must also be explained before publication. In [37], EIS measurements have also been performed on the same material combination. What leads to clearly different IZI 0.01Hz for the global measurements in [37]. How do the authors explain the differences? Does it have something to do with the local measurement?
Response 2: Like the reviewer point out, the |Z|0.01 Hz values were quite different after treatment by T. funiculosus. We thought there are two dominate reasons. First, the results showed in reference [27] exhibited the overall corrosion resistance of the PU coatings after colonized by T. funiculosus, but the results showed in this manuscript were reflected barrier properties of local area of the coatings. On the entire coating area, there were including un-attacked area and attacked area by T. funiculosus because the randomly colonization of it. The approach we selected on reference [27] cannot distinguish these two different areas and the EIS results only can reflect the whole picture. However, the barrier properties of these two different areas on PU coatings were quite different as shown in this manuscript. Actually, the attacked area by T. funiculosus showed poorer corrosion resistance properties and the |Z|0.01 Hz value was lower than the intact area of the PU coatings as shown in Figure 8. Another major reason was that the measurement environment was different. In reference [27], the PU coatings was explored in humid and hot atmospheric environments, and the three-electrodes system was in a 0.05% NaCl solutions. However, the three-electrodes system was in a T. funiculosus-contained PDB medium culture in this manuscript. When the PU coatings degradation in atmospheric environments, the PU coatings was become the only carbon resource (nutrients) to support metabolism of T. funiculosus. For colonization in liquid culture medium, not only the PU coatings but also the carbon resources in PDB medium culture such as potato, dextrose, and peptone can provide nutrients for metabolism of T. funiculosus. Hence, the attack from T. funiculosus for PU coatings could be more severe in atmospheric environments (reference [27]) than in liquid media (this manuscript). It is another reason why the |Z|0.01 Hz value exhibited in reference [27] was lower than that in this manuscript after 14 days of incubation.
In this manuscript, we focused on the differential local degradation of the PU coatings caused by T. funiculosus colonization and its effect on the corrosion resistance of the coatings. Therefore, we did not compare the EIS results in this manuscript, nor did we explain the reasons for the differences. We also pointed out the critical thinking of this manuscript and the different between the reference [27] and this manuscript in the last paragraph of introduction.
Comment 3: Then the caption for figure 8 needs to be changed. Here, for example, (a,d) are the Bode plots and not (a,b).
Response 3: According to the reviewer’s comment, we indeed made a mistake here. We have already revised the caption for figure 8. The caption for Figure 8 could be “Figure 8. The Bode plot (a, d), phase plot (b, e), and Nyquist plot (c, f) of the #85 and #18 electrodes after immersion in T. funiculosus contained medium.”
